# ROA and ROE Forecasting in Iron and Steel Industry Using Machine Learning Techniques for Sustainable Profitability

Mehmet Kayakus [1] , Burçin Tutcu [2,*], Mustafa Terzioglu [2] , Hasan Talaş [2] and Güler Ferhan Ünal Uyar [3]

1. Department of Management Information Systems, Faculty of Social Sciences and Humanities, Akdeniz University, Antalya 07058, Turkey; mehmetkayakus@akdeniz.edu.tr
2. Department of Accounting and Taxation, Korkuteli Vocational School, Akdeniz University, Antalya 07058, Turkey; mterzioglu@akdeniz.edu.tr (M.T.); htalas@akdeniz.edu.tr (H.T.)
3. Department of Business Administration, Faculty of Economics and Administrative Sciences, Akdeniz University, Antalya 07058, Turkey; guleruyar@akdeniz.edu.tr
* Correspondence: burcintutcu@akdeniz.edu.tr

**Abstract:** Return on equity (ROE) and return on assets (ROA) are important indicators that reveal the sustainability of a company's profitability performance for both managers and investors. The correct prediction of these indicators will provide a basis for the strategic decisions made by the company managers. The estimation of these signs is a significant factor in supporting the decisions and up-to-date knowledge of potential investors. In this study, return on equity and return on assets were estimated using artificial neural networks (ANNs), multiple linear regression (MLR), and support vector regression (SVR) on the financial data of thirteen companies operating in the iron and steel sector. The success of predicting ROA in the designed model was 86.4% for ANN, 79.9% for SVR, and 74% for MLR. The success of estimating the ROE of the same model was 85.8% for ANN, 80.9% for SVR, and 63.8% for MLR. It is concluded that ANN and SVR can produce successful prediction results for ROA and ROE both accurately and reasonably.

**Keywords:** ROA; ROE; sustainable profitability; machine learning; iron and steel firms

## 1. Introduction

In the global economy, it is known that one of the most important efforts countries can make to improve their economies is to develop their industries. It is accepted that the driving force behind the industrial development of national economies is the iron and steel sector. The iron and steel sector is important for the national economy since it produces intermediate goods for the manufacturing industry and has a high export potential. Global crude steel production in 2021 was 1.95 billion tons (Mt), increasing by 3.7% compared to 2020. In 2021, crude steel production increased in all countries of the world except for China, Iran, Indonesia, and Malaysia. China nevertheless continues to lead in global steel production, with a production amount of 1 billion, 33 million tons despite experiencing a 3% contraction in production. India is the second largest producer of steel, with a production rate of 17.8% and 118.2 million tons, which increased by 15.8% and 96 million tons compared to 2020. Japan ranked third, with a production of 0.3 million tons [1].

According to the literature, it is possible to reach different studies investigating the impact of working capital on profitability. Wang (2002) evaluated the strength of working capital on profitability by examining the data of 1555 firms in Japan and 379 firms in Taiwan. In this study, the relationship between the cash transition cycle and profitability was examined using the correlation study method, and a negative correlation was reached [2]. In his study, Eljelley (2004) examined the relationship between the profitability and liquidity of a listed company operating in Saudi Arabia. Using the least squares method, a negative relationship was found between liquidity and profitability. Another finding

is that the cashback period is more effective in measuring liquidity when compared to the current ratio [3]. Lazaridis and Tryfonidis (2006) analyzed the data of 131 companies listed on the Athens Stock Exchange for the 2001–2004 period. Return on gross sales and cash turnover were analyzed using regression analysis. They concluded that there is a negative relationship between cash turnover, receivables turnover, inventory turnover, and profitability [4]. Garcia-Teruel and Martinez-Solano (2007) analyzed the effect of working capital on firm performance using data from small and medium-sized firms operating in Spain. The study covers the years 1996–2002, and a total of 8872 data were used. As a result of the analyses, consistent with previous studies, it was found that company managers reduced their inventories and receivables in order to increase the value of the company. In other words, a negative relationship was found between profitability and working capital indicators [5]. Zariyawati et al. (2009) examined the relationship between working capital and firm profitability in Malaysian listed firms in different sectors. In the study, the cash return period was used as an indicator of working capital. Using a total of 1628 data, negative and statistically significant results were obtained between the cash return period and profitability [6]. Sharma and Kumar (2011) examined the effect of working capital management on firm profitability in their study on Indian firms. In the study covering the years 2000–2008, the data of 263 firms were used. As a result of the analysis, it was found that there is a positive relationship between profitability and working capital. In the continuation of the study, the impact of each operating capital element on profitability is analyzed, and it was found that there is a negative relationship between the duration of inventory holding and the average payment period of trade payables and profitability, while there is a positive relationship between the average collection period of receivables and profitability [7]). Mary et al. (2012) researched the effects of the cash conversion period, sales volume, and debt payment period on return on assets using multiple regression analysis. They analyzed the data of five large brewing companies worldwide for the 2000–2011 period. As a result, they emphasized that the cash conversion period, sales volume, and debt service period have significant effects on profitability [8]. Makori and Jagongo (2013) used correlation and regression analysis to assess the effect of the cash conversion cycle, receivable collection, inventory holding, and debt payment period on the return on assets of listed companies on the Nairobi Stock Exchange for the period 2003–2012. A negative relationship was found between the receivable collection period and the cash conversion cycle and profitability, while a positive relationship was found between inventory holding and debt payment period and profitability [9]. Muhammad et al. (2015) used regression analysis to assess the effects of the collection period of receivables, the current ratio, the size of the business, the holding period in stock, and the payment period on the return on assets of food companies traded on the Nigerian Stock Exchange for the 2008–2012 period. There is a negative relationship between profitability and the holding period in stock and the debt payment period and a positive relationship between the current ratio, the size of the enterprise, and the collection period [10]. Postula and Chmielewski (2019) analyzed the correlation between intangible assets and the R&D expenditures, EBITDA level, and market capitalization of 222 publicly traded information and communication technology companies. A fixed effect panel regression model (the panel regression model) was used in the analysis. The findings confirmed a relationship between intangible assets and R&D expenditures and EBITDA level and market value. In addition, this research determined the direction for further research in line with the evaluation of the relationship between sales, assets, and income levels in the analysis of companies' economic and financial situations [11]. Tadić et al. (2019) conducted their research on a sample of two hundred companies in the food industry of the Republic of Serbia. The aim of the study, based on correlation and multiple regression analysis techniques, was to determine whether and to what extent there is a connection between assumed critical success factors and profitability and to examine the contribution of these critical factors to profitability estimation. The study concluded that in the context of the food industry in Serbia, productivity, innovation, quality, and flexibility as critical success

factors were shown to be positively correlated with profitability indicators (ROA, ROE, EBITDA) [12]. Pechlivanidis et al. (2021) evaluated the capability of intangible and other assets to predict corporate profitability. LSTM (long short-term memory) deep learning was used in the evaluation. In this research, the financial data of companies traded in the Greek stock exchange, which prepared their financial reports in accordance with IFRS in the 2000–2018 period, were used. The results of the research confirmed that the LSTM deep learning model improves the corporate profitability estimation of goodwill and intangible assets [13]. Studies in the literature support the existence of a negative relationship between working capital and profitability rates. Many studies in the literature have examined the data of past periods and have determined that one of the most important factors determining managerial success is the management of operating capital. If profitability forecasting can be performed in enterprises, this will enable managers to manage their working capital more effectively. For this purpose, in this study, profitability estimates of enterprises will be made using machine learning methods in order to contribute to the literature. In Table 1, studies on firm valuation and firm performance using machine learning techniques and which are close to the subject of this study are given.

The fact that the iron and steel sector is so large makes the decisions made by the sector's managers much more important. Business managers use historical data when making strategic decisions. However, they cannot predict exactly how these data will affect the forthcoming performance. Our aim in conducting this research is to provide support to the decision-making of information users by estimating the sustainability of the profitability of the companies in the sector. In accordance with this purpose, firstly, a literature review was carried out. The results of this review revealed that previous studies were carried out mostly using correlation and regression analyses. The profitability of the business and the relationship between the calculated ratios are revealed. Although there are statistical models in the literature, studies focusing on the use of machine learning techniques for main industrial sectors such as iron and steel could not be found. For this purpose, it is thought that the present research will contribute to the literature from the perspective of sustainable profitability. In the Section 2, a data set of 13 companies in the iron and steel industry, with complete financial statements selected from the Equity RT database, was selected in accordance with the purpose, and predictions were made using machine learning methods. The findings and results of the research are explained in the Section 4, and the superior aspects of the study will also be emphasized. In the Section 4, which is the Section 4 of the study, the importance and effects of the results of the study are presented.

**Table 1.** Systematic literature review.

| Author/Year | Article Title | Variables | Machine Learning Technique |
|---|---|---|---|
| (Mousa et al., 2022) [14] | Using Machine Learning Methods to Predict Financial Performance: Does Disclosure Tone Matter? | Output: Financial Performance (Earnings per Share) Input: Financial Leverage, Bank Size, Market to Book Ratio, Beta of The Company, Bank Age | Linear Discriminant Analysis, Quadratic Discriminant Analysis, and Random Forest |
| (Zhang et al., 2019) [15] | A Contrastive Study of Machine Learning on Energy Firm Value Prediction | Output: Deal value Input: EBIT, ROE, ROA, CAPEX, M&A Type, Asset Turnover, Cash Debit Ratio, Total Debt to Assets, Firm Type, Nationality, Acquisition Year, Share | Decision Tree Regression, Supported Vector Regression, Artificial Neural Network |
| (Erdal & Karahanoğlu, 2016) [16] | Bagging ensemble models For Bank Profitability: Empirical Research on Turkish Development and Investment Banks | Output: ROE Input: Non-Interest Income/Total Income, Other Revenues/Total Assets, Equity/Total Assets, Loans/Total Assets, Liquid Assets/Total Assets, Non-Performing Loan/Total Loans, Personnel Expenditure/Other Expenditure, Foreign Currency Assets/Total Assets, Total Credits/Total Assets, Net Currency Position/Total Equity | Decision Stump, Reduced Error Pruning Tree, Random Tree |
| (JC et al., 2022) [17] | AI-Based Prediction of Capital Structure: Performance Comparison of ANN SVM and LR Models | Output: Total Debt/Equity Input: Total Liabilities/Equity, Revenues/Cash and Equivalents, Revenues/Current Assets, Revenues/Equity, Net Income/Equity, Gross Margin, EBITDA Margin, Net Income Margin, Current Assets/Current Liabilities, Current Liabilities/Equity, Total Liabilities/Total Assets, EBIT/Interest Expenses | Artificial Neural Network, Support Vector Regression, and Linear Regression |
| (Saberi et al., 2016) [18] | Forecasting the Profitability in the Firms Listed in Tehran Stock Exchange Using Data Envelopment Analysis and Artificial Neural Network | Output: ROA Input: Return on Investment, Capitalization of Exploration Cost, Dupont Ratios | Artificial Neural Network |



**Table 1.** *Cont.*

| Author/Year | Article Title | Variables | Machine Learning Technique |
|---|---|---|---|
| (Skobic et al., 2020) [19] | Machine learning algorithms in the profitability analysis of Casco insurance | Output: Client Profitability<br>Input: Client Age, Client Gender, Discount, Casco claims through history,<br>Car insurance claims through history,<br>Number of Casco policies through history,<br>Casco profit through history,<br>Casco profit for client | Logistic regression, Artificial Neural Network, Decision tree |
| (de Andrés et al., 2004) [20] | The Use of Machine Learning Algorithms for the Study of Business Profitability: A New Approach Based on Preferences | Output: Business Profitability<br>Input: Use of Fixed Capital, Debt Quality, Indebtedness, Short-Term Liquidity, Debt Cost, Share of Labor Costs, Average Sales per Employee, Net Sales | Learning to Assess from Comparison Examples and Recursive Feature Elimination Algorithms |
| (Kuzey et al., 2014) [21] | The Impact of Multinationalism on Firm Value: A Comparative Analysis Of Machine Learning Techniques | Input: Firm Value<br>Output: Asset Structure and Growth Rate, Size, Leverage, Asset Structure and Growth<br>Rate, Sales Growth, Capital Expenditure, Profitability, Liquidity | Artificial Neural Networks and Decision Trees |
| (Zahariev et al., 2022) [22] | Estimation of Bank Profitability Using Vector Error Correction Model and Support Vector Regression | Input: ROE and ROA<br>Output: Inflation and other macroeconomic determinants | Support Vector Regression and Vector Error Correction Model |

## 2. Materials and Methods

In this study, the ROA and ROE values of 13 iron and steel companies were estimated using machine learning methods. Artificial neural networks, support vector regression, and multiple linear regression methods were used in the study. In the models created in the study, there are 5 independent variables and 1 dependent variable.

### 2.1. Data Set

In the study, a data set of 13 companies in the iron and steel industry, with complete financial statements selected from the Equity RT database, was used. Quarterly financial statement data were used in a time range from 2015 Q3 to 2021 Q3. The company names and the countries they operate in are given in Table 2.

**Table 2.** Company set used in the study for the period 2015 Q3-2021 Q3.

| Company Code | Company Name | Operating Country |
|---|---|---|
| BOW PL | Bowim sa | Poland |
| CHMF RUM | Severstal | Russia |
| COG PL | Cognor holding sa | Poland |
| EREGL IS | Eregli demir celik | Turkey |
| FER PL | Ferrum sa | Poland |
| IZS PL | Izostal sa | Poland |
| KRDMA IS | Kardemir | Turkey |
| MAGN RUM | Mmk | Russia |
| OZBAL IS | Ozbal celik boru | Turkey |
| SZR PL | Stalprodukt sa | Poland |
| TRMK RUM | Tmk | Russia |
| URKZ RUM | Uralskaya kuznica | Russia |
| ZRE PL | Zremb-chojnice sa | Poland |

The most important reasons for using these companies as data are that they are listed in the countries in which they operate, and their profitability performance is used as an indicator by potential investors. Potential investors will be able to use this model to guide their decisions. At the same time, the fact that the data of these companies are independently audited increases the reliability of their financial items. Thus, this reliability will contribute to a more reliable interpretation of the model results.

In the study, ROA and ROE estimation models were designed separately for each company in the Table. The variables in these designed models are given in Table 3.

**Table 3.** Variables used in the model.

| Output Variables | Input Variables |
|---|---|
| ROA<br>ROE | Total Assets<br>Current Ratio<br>Leverage Ratio<br>Assets Turnover<br>Working Capital |

As mentioned in the literature section, previous studies have focused on working capital, leverage, and profitability. This focus was taken as the starting point in the establishment of the model [14,16,17,21,22] In this study, a total of 1950 financial data obtained from

the quarterly financial statements of 13 companies were entered into the Knime Program used for machine learning methods.

### 2.2. Artificial Neural Networks

ANNs emerged by modeling the working structure of the human brain. Neurons are used as processing elements in an ANN. Neurons are interconnected through data channels. While each neuron can have multiple inputs, it can only have one output. A neuron's inputs can be from external stimuli or outputs from other neurons [23]

The input layer provides data to the network. The data to be used in the model are transferred through this layer. The hidden layer processes the data received from the input layer and transfers them to the output layer. The output layer also transfers the generated data to the outside [24]

As seen in Figure 1, the data in the input layer ($x_1$, $x_2$, $x_3$, ... , $x_n$) are multiplied by weight values ($w_0$, $w_1$, $w_2$, ... , $w_n$) and transmitted to the next neural cell. The weights indicate the importance and impact of the data coming into the cell. The weights can be randomly assigned at the beginning, or the weights of a model trained in the past can be used as the input-initial weights. However, when assigning these values, they should be determined randomly from positive or negative values. In cases where a value of zero is given, learning will not take place since the calculation will always be the same in the layers. The summation function is used to calculate the net input of the cell [25] As seen in Equation (1), each datum coming into the cell is multiplied by its weight, and the net input to the network is calculated as follows:

$$NET = \sum_{i=1}^{N} x_i w_i \tag{1}$$

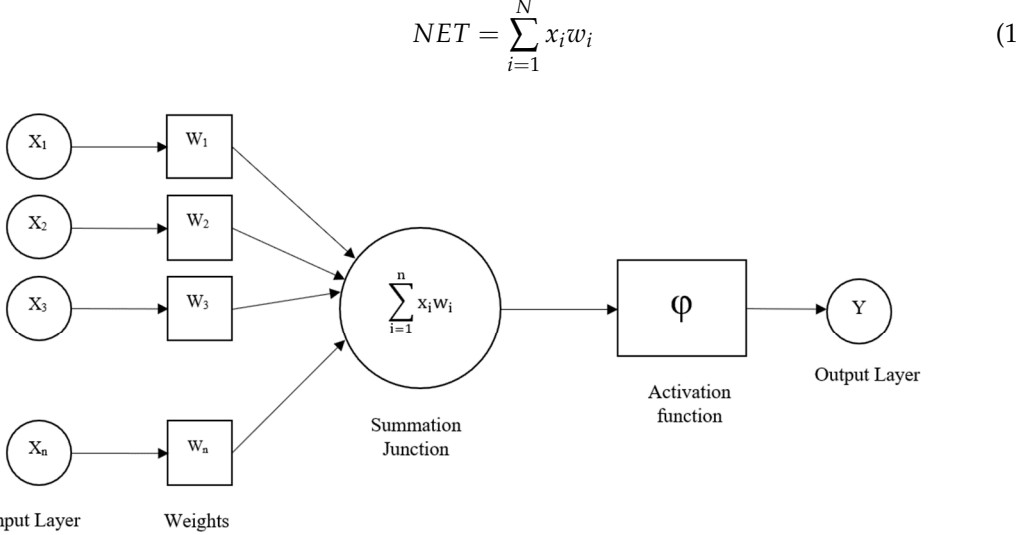

**Figure 1.** ANN structure.

Activation functions determine how the outputs produced in the sum function of neurons should change. The activation function can be selected in two ways. Non-linear activation functions are generally used in artificial neural network models. Non-linear activation functions increase the complexity level of our models. In this way, our ANN models learn the complex structures of data sets in a better way and give more successful results. The activation functions should be continuous and should be able to take first-order derivatives. The Sigmoid function is the most commonly used activation function. It is shown in Equation (2) [26]

$$f(Net) = \frac{1}{1 + e^{-NET}} \tag{2}$$

ANNs are divided into forward and back-propagation networks. In forward propagation networks, data are transferred from the input to the output. In back-propagation networks, the error between the true value and the predicted value is found, and the

weights are updated according to this error. By updating the weight coefficients, the model is optimized [27].

### 2.3. Support Vector Regression

The algorithm is a classification algorithm based on the determination of the hyperplane that separates two or more classes. An infinite number of planes can be determined to separate the classes from each other. The aim of the support vector machine (SVM) algorithm is to determine a hyperplane that separates the classes from each other and where the distances of the class samples to the hyperplane are maximized. An infinite number of planes can be determined to separate the classes from each other. The aim of the support vector machine algorithm is to determine a hyperplane that separates the classes from each other and where the distances of the class samples to the hyperplane are maximized [28].

The main aim of regression is to determine the line or curve that can fit the maximum point in a margin range with the smallest error [29]). In Figure 2, the value that expresses the distance of the line from the dashed line is called the epsilon ($\epsilon$) distance. $\xi_i$ and $\xi_i^*$ values are outliers. Here, the regression curve is determined by using the $\epsilon$ and $\xi_i$ and $\xi_i^*$ values.

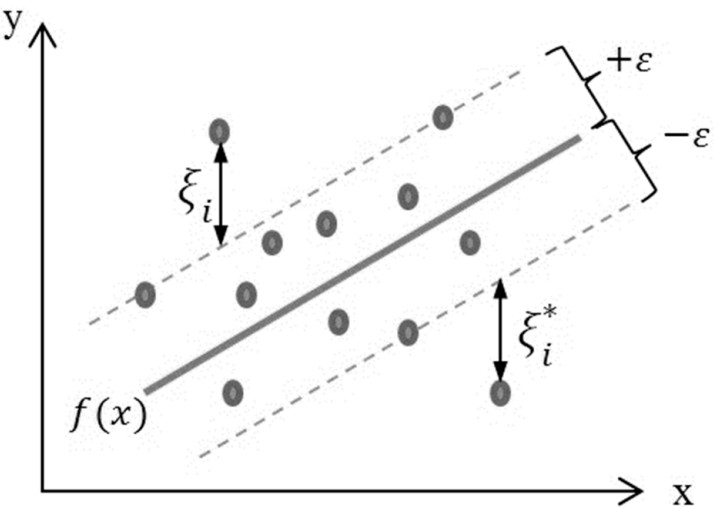

**Figure 2.** Structure of SVM.

Equation (3) is used to find a regression function in SVR hyperspace:

$$f(x) = w^T \varnothing(x) + b \tag{3}$$

### 2.4. Multiple Linear Regression

In regression analysis, a dependent variable and one or more independent variables are used to obtain a mathematical model. If there is one independent variable in the model to be found, the method is expressed by a simple linear regression. The method used to explain the cause-and-effect relationships between two or more independent variables affecting a variable in a linear model and to determine the effects of these independent variables is called multiple linear regression (MLR) analysis [30] Simple linear regression can be expressed as follows:

$$y = a + bx + e \tag{4}$$

where $y$ is the dependent variable, $a$ is the regression constant, $b$ is the independent variable coefficient, $x$ is the independent variable, and e is the error.

MLR is used when two or more independent variables affect the outcome [31] and can be expressed as follows:

$$y = a + b_1 x_1 + b_2 x_2 + b_3 x_3 + \ldots + b_n x_n + e \tag{5}$$

where $n$ is the number of independent variables, and $b$ is the coefficient of each independent variable.

## 3. Result and Discussion

By pre-processing the variables in the data set, training is made more efficient. At this stage, the noisy and erroneous data are removed, and the data are normalized. There may be inconsistent, inaccurate, and incomplete data in the data set. At this stage, these data are cleaned, and the data set is prepared for training. Since the sizes of the variables in the data set are different, the data set should be normalized so that these values are expressed in the same value range. Normalization will reduce the training time of the models and increase their performance. Different techniques can be used for normalization. These can be listed as rules such as the Min rule, Max rule, median, Sigmoid, and Z-Score. In this study, the data were normalized using the min–max method. This method involved the assignment of variables to values between 0 and 1. The largest and smallest values in a group of data are considered. All other data are normalized according to these values. The aim here is to normalize the smallest value as 0 and the largest value as 1 and to spread all other data to this 0–1 range [32] The min–max normalization equation is as follows:

$$x' = \frac{x_i - x_{min}}{x_{max} - x_{min}} \tag{6}$$

Here, $x'$ is the normalized data, $x_i$ is the input value, $x_{max}$ is the largest number, and $x_{min}$ is the smallest number.

In supervised machine learning, the data set is divided into the training set and the test set. Modeling and training are performed on the training set. The model learns here. Then, the relevant model is tested using the test set, and it is determined how successfully the model learns. In this way, the success of the two sets is evaluated. In the study, 70% of the data set was used for training and 30% for testing. The separation of the data was performed using the linear sampling method.

MSE (mean squared error), RMSE (root mean square error), and $R^2$ (coefficient of determination) statistical metrics were used to evaluate the study methods.

$R^2$ shows the total variation in the regression line we draw according to the actual table. In other words, $R^2$ gives the rate at which multiple independent variables explain the dependent variable. If the $R^2$, which varies between 0 and 1, approaches 0, it is understood that the data are not suitable for the model; that is, the model does not explain the data. The best $R^2$ value is 1 [33] The $R^2$ equation is given as follows:

$$R^2 = 1 - \frac{\text{Unexplained Variation}}{\text{Total Variation}} \tag{7}$$

The MSE is a measure of evaluating success. Especially when using optimization methods, it is an optimizing function that tries to build better models. Success measurement metrics used in regression models provide the opportunity to evaluate the differences between actual values and predicted values. The MSE is the mean squared loss per sample over the entire dataset. To calculate the MSE, the errors in the data between the predicted value and the true value are summed and then divided by the number of data. Models with MSE values close to zero have fewer errors and are considered more successful [34] The MSE equation is given as follows:

$$\text{MSE} = \frac{1}{n} \sum_{j=1}^{n} e_j^2 \tag{8}$$

where $n$ is the number of samples and $e$ is the difference between the actual data and the predicted data, that is, the error value.

To calculate the RMSE, the squares of the differences between the actual value and the prediction obtained from the model are averaged, and the square root of this value is calculated. This provides information on how far the predicted values and the actual observations are from each other. Since it is a measurement of error, keeping this value as low as possible shows that the prediction ability is strong [35] The RMSE equation is given as follows:

$$\text{RMSE} = \sqrt{\frac{\sum_{j=1}^{n} e_j^2}{n}} \tag{9}$$

In this study, a feedback neural network model was designed. Since there is no rule for determining the parameters of the model in the training phase, the trial and error method was used. In the training phase, the best result was obtained in four hidden layers. There are four neurons in each hidden layer. One thousand iterations were performed.

A non-linear support vector regression was used in the study. The radial basis function (RBF) was chosen for the kernel function, which takes the low-dimensional input space and transforms it into a higher-dimensional space. The performance of the SVR is mainly affected by the determination of the C parameter and the kernel parameter. In this study, the C parameter is set to 100, and the kernel parameter is set to 0.1.

In the multiple linear regression method, there are two or more independent variables. The effects of the variables on the system may be different from each other, and these effects may result in positive or negative results. There are several methods for determining the effect ratios of variables in the model. It is one of the step-by-step comparison methods in the backward elimination method. First of all, a significance value is determined. The variable with the current highest $p$-value (probability value) is determined, and if P > SL, the variable is removed from the system, and the model is rebuilt; then, this step is repeated. In order for the backward elimination stage to be finalized, the P < SL condition must be met for all variables. When this condition is met, the next stage is started. The forward elimination method includes the same steps as the backward elimination method. As a difference, the algorithm includes a single variable at the beginning, as opposed to backward elimination, and the number of variables increases with each step. The $p$-value used in these definitions is the probability that, under a given statistical model, the statistical sum of the data is equal to or greater than the observed value. In the study, the $p$-values of each independent variable forming the model are below 0.05.

There are statistically significant differences in the performance results of the three machine learning methods in terms of MSE, RMSE, and $R^2$.

The ideal value for $R^2$ is 1 (%100). This means that the independent variables are strong in explaining the dependent variable. In this study, the $R^2$ was 86.4% for the mean ANN, 79.9% for the SVR, and 74% for the MLR, and these were found to be acceptable values. The ideal value for the MSE is zero. Therefore, models close to zero are more successful. In this study, the MSE of the mean ANN was 0.010, which was found to be closer to the ideal value than the MSEs of 0.036 for SVR and 0.013 for MLR. Moreover, the smaller the RMSE, the better the model. The RMSE was 0.089 for ANN, 0.154 for SVR, and 0.013 for MLR. It was observed that the RMSEs of all three models were close to zero, which is the desired value. According to the results in Table 4, the order of success and error is ANN, SVR, and MLR.

The $R^2$ was 85.8% for ANN, 80.9% for SVR, and 63.8% for MLR, and the results were acceptable for ANN and SVR. It was observed that the success rate was lower for MLR. The MSE results of the study were 0.008 for the ANN, 0.562 for SVR, and 0.146 for MLR, which were found to be close to the ideal. It was seen that the RMSE was 0.085 for ANN, 0.665 for SVR, and 0.320 for MLR, which was also close to the ideal. According to the results in Table 5, the order of success and error is ANN, SVR, and MLR.

**Table 4.** Comparison of models according to their ROA.

| Company Code | ANN | | | SVR | | | MLR | | |
|---|---|---|---|---|---|---|---|---|---|
| | MSE | RMSE | $R^2$ | MSE | RMSE | $R^2$ | MSE | RMSE | $R^2$ |
| BOW PL | 0.004 | 0.064 | 0.921 | 0.001 | 0.030 | 0.887 | 0.001 | 0.009 | 0.991 |
| CHMF RUM | 0.011 | 0.104 | 0.797 | 0.003 | 0.053 | 0.698 | 0.001 | 0.038 | 0.878 |
| COG PL | 0.001 | 0.031 | 0.981 | 0.040 | 0.199 | 0.840 | 0.015 | 0.124 | 0.833 |
| EREGL IS | 0.014 | 0.119 | 0.835 | 0.019 | 0.137 | 0.750 | 0.001 | 0.030 | 0.506 |
| FER PL | 0.010 | 0.099 | 0.911 | 0.146 | 0.383 | 0.834 | 0.012 | 0.108 | 0.309 |
| IZS PL | 0.014 | 0.118 | 0.811 | 0.052 | 0.228 | 0.776 | 0.015 | 0.121 | 0.694 |
| KRDMA IS | 0.017 | 0.131 | 0.698 | 0.010 | 0.098 | 0.737 | 0.005 | 0.074 | 0.711 |
| MAGN RUM | 0 | 0 | 0.996 | 0.046 | 0.214 | 0.729 | 0.012 | 0.110 | 0.887 |
| OZBAL IS | 0.015 | 0.123 | 0.805 | 0.001 | 0.035 | 0.778 | 0.025 | 0.159 | 0.69 |
| SZR PL | 0.004 | 0.065 | 0.943 | 0.074 | 0.272 | 0.809 | 0.038 | 0.195 | 0.716 |
| TRMK RUM | 0.029 | 0.170 | 0.626 | 0.070 | 0.265 | 0.917 | 0.016 | 0.126 | 0.84 |
| URKZ RUM | 0.003 | 0.051 | 0.966 | 0.001 | 0.019 | 0.722 | 0.016 | 0.126 | 0.781 |
| ZRE PL | 0.005 | 0.067 | 0.941 | 0.005 | 0.069 | 0.910 | 0.013 | 0.114 | 0.789 |
| Average | 0.010 | 0.089 | 0.864 | 0.036 | 0.154 | 0.799 | 0.013 | 0.103 | 0.740 |

**Table 5.** Comparison of models according to ROE.

| Company Code | ANN | | | SVR | | | MLR | | |
|---|---|---|---|---|---|---|---|---|---|
| | MSE | RMSE | $R^2$ | MSE | RMSE | $R^2$ | MSE | RMSE | $R^2$ |
| BOW PL | 0.004 | 0.066 | 0.919 | 0.034 | 0.185 | 0.974 | 0.148 | 0.384 | 0.513 |
| CHMF RUM | 0.004 | 0.062 | 0.916 | 0.527 | 0.726 | 0.681 | 0.065 | 0.255 | 0.699 |
| COG PL | 0.003 | 0.053 | 0.947 | 0.212 | 0.461 | 0.738 | 0.110 | 0.331 | 0.458 |
| EREGL IS | 0.015 | 0.121 | 0.833 | 0.252 | 0.502 | 0.716 | 0.079 | 0.28 | 0.880 |
| FER PL | 0.005 | 0.069 | 0.883 | 1.040 | 1.020 | 0.865 | 0.179 | 0.423 | 0.963 |
| IZS PL | 0.008 | 0.087 | 0.929 | 0.068 | 0.260 | 0.875 | 0.205 | 0.453 | 0.629 |
| KRDMA IS | 0.017 | 0.132 | 0.628 | 1.822 | 1.350 | 0.992 | 0.045 | 0.212 | 0.575 |
| MAGN RUM | 0.003 | 0.050 | 0.95 | 0.360 | 0.600 | 0.761 | 0.510 | 0.226 | 0.909 |
| OZBAL IS | 0.008 | 0.090 | 0.816 | 1.671 | 1.293 | 0.900 | 0.138 | 0.371 | 0.804 |
| SZR PL | 0.007 | 0.081 | 0.904 | 0.494 | 0.703 | 0.638 | 0.187 | 0.432 | 0.659 |
| TRMK RUM | 0.010 | 0.099 | 0.865 | 0.203 | 0.450 | 0.802 | 0.057 | 0.238 | 0.305 |
| URKZ RUM | 0.005 | 0.068 | 0.937 | 0.190 | 0.436 | 0.871 | 0.149 | 0.387 | 0.524 |
| ZRE PL | 0.017 | 0.129 | 0.627 | 0.437 | 0.661 | 0.701 | 0.028 | 0.167 | 0.377 |
| Average | 0.008 | 0.085 | 0.858 | 0.562 | 0.665 | 0.809 | 0.146 | 0.320 | 0.638 |

According to the results obtained by the designed model using machine learning techniques, it is believed that managerial decision makers' interpretation of the model using ANN will contribute the most to the sustainability of profitability. At the same time, investors who expect dividends for many years can also apply the ANN technique to this model concerning the sustainability of profitability.

## 4. Conclusions

The iron and steel sectors are one of the leading sectors in the industrialization of many countries. This sector is the main supplier of many other sectors with the intermediate goods it produces, especially in the manufacturing industry and construction sectors. For this reason, the success of iron and steel companies in terms of management significantly affects the success of other sectors. At the same time, companies in this sector are companies that are carefully monitored by investors in the stock exchanges of developed and developing countries alike. The performances of these companies give clues in terms of the economic development of the countries in which they operate, as the iron and steel sector is the main branch of industry. In addition, investors consider the ROA and ROE results while making their stock investments in these companies and include the stocks of these companies in their portfolios according to these results. ROA and ROE are two of the most important metrics revealing the success criteria of these companies, both from a managerial and an investor perspective. In this respect, we believe that these two indicators are especially important for firms to achieve sustainable profitability performance.

In this study, machine learning techniques such as ANN, SVR, and MLR were used to try to reveal the power of predicting the ROA and ROE based on the outcomes of the liquidity, operating capital, and asset management of the enterprise. In the study, a model was created using variables that are frequently cited in the literature, especially as they affect ROA and ROE. This model was applied separately to thirteen companies operating in this sector in Poland, Russia, and Turkey. According to the results obtained, the success of this model in predicting ROA is 86.4% for ANN, 79.9% for SVR, and 74% for MLR. The ROE of the same model was 85.8% for ANN, 80.9% for SVR, and 63.8% for MLR. These findings show that ANN and SVR are both accurate and reasonably capable of producing understandable prediction results for ROA and ROE.

The results show that this model is a viable model for managers in the iron and steel sector to use to evaluate their company's performance and to create future policies while preparing operating budgets. At the same time, securities market investors can diversify the stocks in the iron and steel sector in their portfolios by using the ANN and SVR in this model.

The model designed in this study has been successfully applied in iron and steel sector enterprises. In addition, it is thought that it can be successfully applied to similar heavy industry production sectors (e.g., the machinery industry, mining, metallurgy, energy, defense industry, chemistry, etc.) since the selected independent variables are generally valid in these sectors. It is also predicted that the success of the predictions obtained by the machine learning techniques used will show high performance in future studies on the prediction of important financial ratios such as the ROA and ROE.

**Author Contributions:** Methodology, M.K., B.T. and G.F.Ü.U.; Formal analysis, M.K.; Data curation, M.T. and H.T.; Writing—original draft, M.K., B.T., M.T., H.T. and G.F.Ü.U.; Writing—review and editing, B.T., M.T., H.T. and G.F.Ü.U. All authors have read and agreed to the published version of the manuscript.

**Funding:** This research received no external funding.

**Data Availability Statement:** The data presented in this study are available on request from the corresponding author.

**Conflicts of Interest:** The authors declare no conflict of interest.

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
