# Peer review of "ROA and ROE Forecasting in Iron and Steel Industry Using Machine Learning Techniques for Sustainable Profitability"

_sustainability, doi:10.3390/su15097389_

Round 1
Reviewer 1 Report
Name of the Paper: ROA and ROE Forecasting in Iron and Steel Industry Using Machine Learning Techniques for Sustainable Profitability
The paper provides the prediction of return on equity (ROE)and return on assets (ROA) using artificial neural networks (ANN), multiple linear regression (MLR), and support vector regression (SVR) on the financial data of thirteen companies operating in the iron and steel sector. Authors could predict ROA with different accuracy levels: 86.4% for ANN, 79.9% for SVR, and 74% for MLR. Authors could predict ROE with 85.8% for ANN, 80.9% for SVR, and 63.8% for MLR. They further conclude that ANN and SVR can produce successful prediction results for ROA and ROE, both accurately and reasonably.
General Observation:
1) Please refer to the abstract, “From a, the estimation of these signs will have a significant factor in supporting the decisions of up-to-date and potential investors.” Is unclear.
2) Table 1 is not used in the text.
3) The literature review is very vaguely provided and does not corroborate the research topic.
4) Please refer to “Table 1. In literature, studies on firm valuation and firm performance, which are close to the subject of the study, are given by using machine learning techniques. ‘Table heading may be curtailed to simplify it for better understanding.
5) Please refer to Table 3. Variables Used in the Model. The Table may be bifurcated into Output and Input variables for clarity.
6) Authors should provide a citation to these used variables for their usage in prediction.
7) The manuscript needs thorough revisions as many statements are vague, for instance:
8) Line no.138: “The most important reason for using these companies as data is that they are listed on the stock exchange in the countries in which they operate.”
9) Line no.147: “In this study, 150 financial data for each company obtained from the financial statements, a total of 1950 financial data were entered into the Knime Program used for machine learning methods.”
10) The manuscript needs good formatting to place all equations systematically with their numbers for better readability.
11) Please refer to ‘Table 4. Comparison of Models According to Their ROA.” The table header misses the first column. The company code or name should be unique in all tables i.e. Table 2, Table 4, and Table 5
12) Please refer to line no.275: "Elimination is terminated when P<SL for all variables.’ is unclear.
13) Please refer to line no.280" The p values of the independent variables are below 0.05." is unclear.
14) Please refer to line no.285: The desired value for R2 is 1. “ R2” or “R2” should be the same throughout in manuscript
15) Please refer to line no.291: "RMSE was 0.089 for ANN, 0.154 for SVR, and 0.013 for MLR, which was close to the ideal value.” the RMSE is 0.103
16) Please refer to lines 291 and 299: "which was close to the ideal." is unclear.
17) Authors may provide managerial implications and limitations to these studies.
18) Authors may provide future research directions.
Quality English language may be improved:
Please refer to the abstract, “From a, the estimation of these signs will have a significant factor in supporting the decisions of up-to-date and potential investors.” Is unclear.
Line no.138: “The most important reason for using these companies as data is that they are listed on the stock exchange in the countries in which they operate.”
Line no.147: “In this study, 150 financial data for each company obtained from the financial statements, a total of 1950 financial data were entered into the Knime Program used for machine learning methods.”
Author Response
Response letter to the Comments of Editor and Reviewers on sustainability-2353632
Title of the paper: ROA and ROE Forecasting in Iron and Steel Industry Using Machine Learning Techniques for Sustainable Profitability
First of all, authors would like to thank all the anonymous Reviewers and the Editor again for their efforts and valuable time to review and improve our paper.
Taking into account their constructive suggestions and comments, the paper has been carefully revised following the referees’ comments.

Reviewer 2 Report
The authors of the paper conducted a study to predict ROA and ROE using ANN, SVR, and MLR on financial datatubes of 13 companies in the steel industry. The contribution of this study is limited in terms of computer science, but I think it is meaningful in terms of the problems it raises and the hypotheses it tests.
I would like to see the actual author and year in the author/year column in Table 1 so that we can get more details.
Author Response

(The authors gave the same response as above.)

Reviewer 3 Report
1- Please combine the sections1 and 2 as one section. In this form, the paper is too long. This paper is technical paper not a review paper.
2- Pleas list the sections of paper in the end of introduction.
3- There are many machine learning models, why you choose these three models?
4- How did you select the best values for the hyperparameters of the models?
5- You just use three criteria (i.e., RMSE, R2, and MSE) to evaluate the performance of models! Please use more evaluation criteria parameters. Please check this papers: https://doi.org/10.1007/s11269-023-03497-x; https://doi.org/10.1007/s00521-021-06199-w.
6- More results should be presented.
Dear Editor
In my opinion it can be consider to publish after revising accordingly above comments.
Regards
Author Response

(The authors gave the same response as above.)

Round 2
Reviewer 1 Report
Thank you for the manuscript modification and English revision.